# Spatial-Spectral Network for Hyperspectral Image Classification: A 3-D CNN and Bi-LSTM Framework

**Junru Yin \*, Changsheng Qi, Qiqiang Chen and Jiantao Qu**

College of Computer and Communication Engineering, Zhengzhou University of Light Industry, Zhengzhou 450000, China; qics@zzuli.edu.cn (C.Q.); chenqq@zzuli.edu.cn (Q.C.); qujt@email.zzuli.edu.cn (J.Q.)
**\*** Correspondence: yinjr@zzuli.edu.cn; Tel.: +86-132-038-20152

**Abstract:** Recently, deep learning methods based on the combination of spatial and spectral features have been successfully applied in hyperspectral image (HSI) classification. To improve the utilization of the spatial and spectral information from the HSI, this paper proposes a unified network framework using a three-dimensional convolutional neural network (3-D CNN) and a band grouping-based bidirectional long short-term memory (Bi-LSTM) network for HSI classification. In the framework, extracting spectral features is regarded as a procedure of processing sequence data, and the Bi-LSTM network acts as the spectral feature extractor of the unified network to fully exploit the close relationships between spectral bands. The 3-D CNN has a unique advantage in processing the 3-D data; therefore, it is used as the spatial-spectral feature extractor in this unified network. Finally, in order to optimize the parameters of both feature extractors simultaneously, the Bi-LSTM and 3-D CNN share a loss function to form a unified network. To evaluate the performance of the proposed framework, three datasets were tested for HSI classification. The results demonstrate that the performance of the proposed method is better than the current state-of-the-art HSI classification methods.

**Keywords:** 3-D convolutional neural network (3-D CNN); deep learning; bidirectional short-term memory (Bi-LSTM); hyperspectral images (HSIs) classification; spatial-spectral

## 1. Introduction

With the rising potential of remote-sensing applications in real life, research in remote-sensing analysis is increasingly necessary [1,2]. Hyperspectral imaging is commonly used in remote sensing. A hyperspectral image (HSI) is obtained by collecting tens or hundreds of spectrum bands in an identical region of the Earth's surface by an imaging spectrometer [3,4]. In an HSI, each pixel in the scene includes a sequential spectrum, which can be analyzed by its reflectance or emissivity to identify the type of material in each pixel [5,6]. Owing to the subtle differences among HSI spectra, HSIs have been applied in many fields. For instance, hydrological science [7], ecological science [8,9], geological science [10,11], precision agriculture [12,13], and military applications [14].

In recent decades, the classification of HSIs has become a popular field of research for the hyperspectral community. While the abundant spectral information is useful for improving classification accuracy compared to natural images, the high dimensionality presents new difficulties [15,16]. The HSI classification task has the following challenges: (1) HSI has high intra-class variability and inter-class diversity. These are influenced by many factors, such as changes in lighting, environment, atmosphere, and temporal conditions. (2) The available training samples are limited in relation to the high dimensionality of HSIs. As the dimension of HSIs increases, the required training samples also keep increasing, while the available samples of HSIs are limited. Therefore, these factors can result in an unsuitable methodology, reducing the classifier's ability for generalization.

In early HSI classification studies, most approaches focused on the influence of HSI spectral features on classification results. Therefore, several existing methods are based on pixel-level HSI classification, for instance, multinomial logistic regression [17], support

vector machines (SVM) [18–20], K-nearest neighbor (KNN) [21], neural networks [22], linear discriminative analysis [23–25], and maximum likelihood methods [26]. SVM is mainly dedicated to the transformation of linearly inseparable problems into linearly separable problems by finding the optimal hyperplane (such as the radial basis kernel and composite kernel [19]), which finally completes the classification task. Since these methods utilize the spatial context information insufficiently, the classification results obtained by these pixel classifiers using only spectral features are unsatisfactory. Recently, researchers have found that spatial feature-based classification methods have significantly improved the representation of hyperspectral data and classification accuracy [27,28]. Thus, more researchers are combining spectral-spatial features into pixel classifiers to exploit the information of HSIs completely and improve the classification results. For example, multiple kernel learning uses various kernel functions to extract different features separately, which are fed into the classifier to generate a map of classification results. In addition, researchers in [29,30] segmented HSIs into multiple superpixels to obtain similar spatial pixels based on intensity or texture similarity. Although these methods have achieved sufficient performance, hand-crafted filters extract limited features, and most can only extract shallow features. The hand-crafted features depend on the expert's experience in setting parameters, which limits the development and applicability of these methods. Therefore, for HSI classification, the extraction of deeper and more easily discernible features is the key.

In recent decades, deep learning [31–33] has been extensively adopted in computer vision, for instance, in image classification [34–36], object detection [37–40], natural language processing [41], and has obtained remarkable performance in HSI classification. In contrast to traditional algorithms, deep learning extracts deep information from input data through a range of hierarchical structures. In detail, some simple line and shape features can be extracted at shallow layers, while deeper layers can extract abstract and complex features. The deep learning process is fully automatic without human intervention and can extract different feature types depending on the network; therefore, deep learning methods are suitable for handling various situations.

At present, there are various deep-learning-based approaches for HSI classification, including deep belief networks (DBNs) [42], stacked auto-encoders (SAEs) [43], recurrent neural networks (RNNs) [44,45], convolutional neural networks (CNNs) [46,47], residual networks [48], and generative adversarial networks (GANs) [49]. The SAEs consist of multiple auto-encoder (AE) units that use the output of one layer as input to subsequent layers. Li et al. [50] used active learning techniques to enhance the parameter training of SAEs. Guo et al. [51] reduce the dimensionality by fusing principal component analysis (PCA) and kernel PCA to optimize the standard training process of DBNs. Although these methods have adequate classification performance, the number of model parameters is large. In addition, the HSI cube data are vectorized, and the spatial structure can be corrupted, which leads to inaccurate classification.

The CNN can extract local two-dimensional (2-D) spatial features of images, and the weight-sharing mechanism of a CNN can effectively decrease the number of network parameters. Therefore, CNNs are widely used in HSI classification. Hu et al. [52] proposed a deep CNN with five one-dimensional (1-D) layers, which receives pixel vectors as input data and classifies HSI data in the spectral domain only. However, this method loses spatial information, and the network depth is shallow, limiting the extraction of complex features. Zhao et al. [53] proposed a CNN2D architecture, in which multi-scale, convolutional AEs based on Laplace pyramids obtain a series of deep spatial features, while the PCA extracts three principal components. Then, logistic regression is used as a classifier that connects the extracted spatial features and spectral information. However, the method does not consider spectral features and the classification effect on improvement. To extract the spatial–spectral information, Chen et al. [54] proposed three convolutional models for creating input blocks of their CNN3D model using full-pixel vectors from the original HSI. This method extracts spectral, spatial, and spatial–spectral features, which generate data

redundancy. In addition, Liu et al. [55] proposed a bidirectional-convolutional long short-term memory (Bi-CLSTM) network with which the convolutional operators across spatial domains are combined into a bidirectional long short-term memory (Bi-LSTM) network to obtain spatial features while fully incorporating spectral contextual information.

In summary, sufficiently exploiting features of HSI data and minimizing computational burden are the keys to HSI classification. This paper proposes a joint unified network operating in the spatial–spectral domain for the HSI classification. The network uses three layers of 3-D convolution for extracting the spatial–spectral feature of HSI, and subsequently adds a layer of 2-D convolution to further extract spatial features. For spectral feature extraction, this network treats all spectral bands as a sequence of images and enhances the interactions between spectral bands using Bi-LSTM. Finally, two fully connected (FC) layers are combined and use the softmax function for classification, which forms a unified neural network. We list the major contributions of our proposed method.

1.  A Bi-LSTM framework based on band grouping is proposed for extracting spectral features. Bi-LSTM can obtain better performance in learning contextual features between adjacent spectral bands. In contrast to the general recurrent neural network, this framework can better adapt to a deeper network for HSI classification.
2.  The proposed method adopts 3-D CNN for extracting the spatial–spectral features. To reduce the computational complexity of the whole framework, PCA is used before the convolutional layer of the 3-D CNN to reduce the data dimensionality.
3.  A unified framework named the Bi-LSTM-CNN is proposed which integrates two subnetworks into a unified network by sharing the loss function. In addition, the framework adds the auxiliary loss function, which balances the effects of spectral and spatial-spectral features for the classification results to increase the classification accuracy.

The structure of the remaining part is as follows. Section 2 describes long short-term memory (LSTM), a 3-D CNN, and the framework of the Bi-LSTM-CNN. Section 3 introduces the HSI datasets, experimental configuration, and experimental results. Section 4 provides a detailed analysis and interpretation of the experimental results. Finally, conclusions are summarized in Section 5.

## 2. Materials and Methods

### 2.1. Related Work

#### 2.1.1. LSTM

Some tasks need to consider the information of previous and subsequent inputs in processing the current input. RNNs can solve these problems and handle the spectral contextual information of an HSI. Figure 1 shows the architecture of an RNN. Given a series of values $x^{(1)}, x^{(2)}, \ldots, x^{(t)}$ as input data, the formula for each cell structure in the RNN network is shown as Equations (1) and (2):

$$h^{(t)} = \tan h \left( W h^{(t-1)} + U x^{(t)} + b_a \right), \tag{1}$$

$$O^{(t)} = V h^{(t)} + b_o, \tag{2}$$

where $W$, $U$, $V$ denote the weight matrices that represent the relation of two nodes. In detail, $W$ connects the previously hidden node and the currently hidden node, $U$ connects the input node and the hidden node, and $V$ connects the hidden node and the output node. Vectors $b_a$ and $b_o$ are bias vectors. At time $t$, $x^{(t)}$ represents the input value, $h^{(t)}$ represents the hidden value, and $O^{(t)}$ represents the output value. The tanh is a nonlinear activation function. The initialization value of $h^{(0)}$ in Equation (1) is set to zero. Equation (1) indicates that the output is jointly determined by the input $x^{(t)}$ at time $t$ and the $h^{(t-1)}$ at time $t-1$. As $|W| < 1$ or $|W| > 1$, $h^{(t)}$ will be closer to infinity or zero as time increases. This will cause the gradient to disappear or explode in the backpropagation phase. In other words,

when the relevant information is very far from the current location, RNN will not utilize this information effectively. RNN cannot solve the problem of long-term dependence.

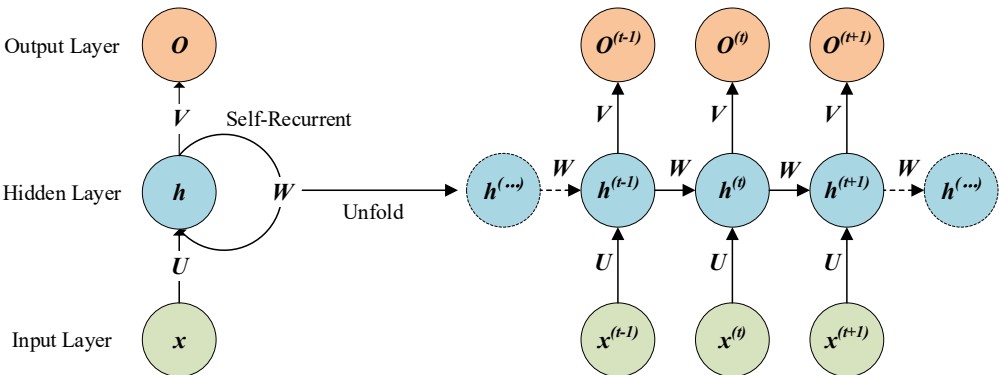

**Figure 1.** RNN architecture.

The LSTM network is proposed to solve this problem. Through the gating mechanism, the LSTM not only remembers past information but also filters some unimportant information. The LSTM is effective in solving the long dependency problem in the RNNs.

The architecture of LSTM is shown in Figure 2. The memory cell is a critical component of the LSTM, replacing the hidden unit of the RNNs. The cell state runs throughout the cell, but it has few branches to ensure information flows unchanged throughout the RNNs. The LSTM network has a structure called a gate, which can delete or add information about the cell state. The gate is combined by the Hadamard product operation and the sigmoid function and can filter which information is allowed to pass. The LSTM has three gates: the input gate, which determines how the previous memory is combined with the new input information; the output gate, which controls if the state of the cell at the next time step will affect other neurons; and the forget gate, which regulates the cell state, causing the cell to forget or remember a previous state. The candidate cell value stores updated information from the output of the input gate operation. At time $t$, the forward propagation of the LSTM is defined as Equations (3)–(8).

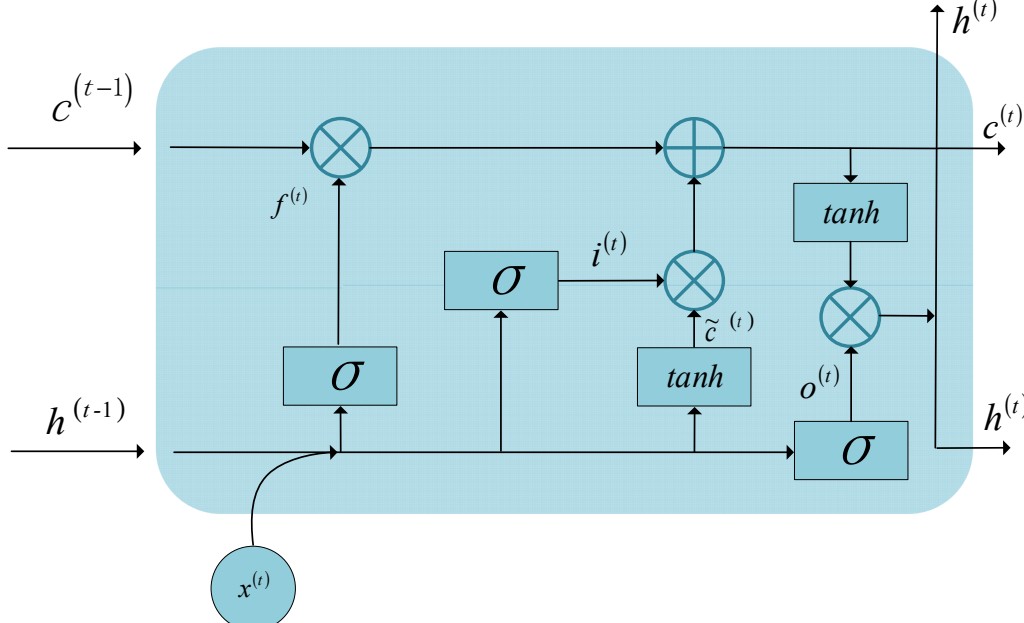

**Figure 2.** LSTM architecture.

Input gate:

$$i^{(t)} = \sigma\left(W_i x^{(t)} + U_i h^{(t-1)} + b_i\right), \tag{3}$$

Forget gate

$$f^{(t)} = \sigma\left(W_f x^{(t)} + U_f h^{(t-1)} + b_f\right), \tag{4}$$

Output gate

$$o^{(t)} = \sigma\left(W_o x^{(t)} + U_0 h^{(t-1)} + b_o\right), \tag{5}$$

Candidate cell value

$$\widetilde{c}^{(t)} = \tan h\left(W_c x^{(t)} + U_c h^{(t-1)} + b_c\right), \tag{6}$$

Cell state

$$c^{(t)} = i^{(t)} * \widetilde{c}^{(t-1)} + f^{(t)} * c^{(t-1)}, \tag{7}$$

LSTM output

$$h^{(t)} = o^{(t)} * \tan h\left(c^{(t)}\right), \tag{8}$$

where $\sigma$ denotes the logistic sigmoid function and $*$ represents the Hadamard product operation. The matrices $W_i$, $W_f$, $W_o$, $W_c$, $U_i$, $U_f$, $U_o$, and $U_c$ are weight matrices. The vectors $b_i$, $b_f$, $b_o$, and $b_c$ are bias vectors.

### 2.1.2. CNN

CNNs are being applied with great success in many research areas. A CNN can extract various kinds of features from an image, such as color, texture, shape, and topology, so it has the advantage of processing 2-D images, such as identifying displacement, scaling, and other forms of distortion invariance. Similar to biological neural networks, the structure of the weight-sharing network of CNNs decreases the number of parameters, thus decreasing the complexity of the network model.

CNNs include 1-D CNN, 2-D CNN, and 3-D CNN. The 1-D CNN is mainly adopted for sequence data processing; the 2-D CNN is usually adopted for image recognition; the 3-D CNN is mainly used for medical image and video data recognition. A CNN consists of three structures: convolution layer, activation function, and pooling layer. There are no pooling layers in some CNNs. In detail, the purpose of the convolutional layer is for the extraction of the input data features; with more convolutional layers, the extracted features are more complex. The activation function increases the nonlinearity of the neural network model. The pooling layer preserves the main features while decreasing the number of parameters and calculations, preventing overfitting and improving model generalization. The schematic diagrams of the 2-D convolution and 3-D convolution are shown in Figure 3.

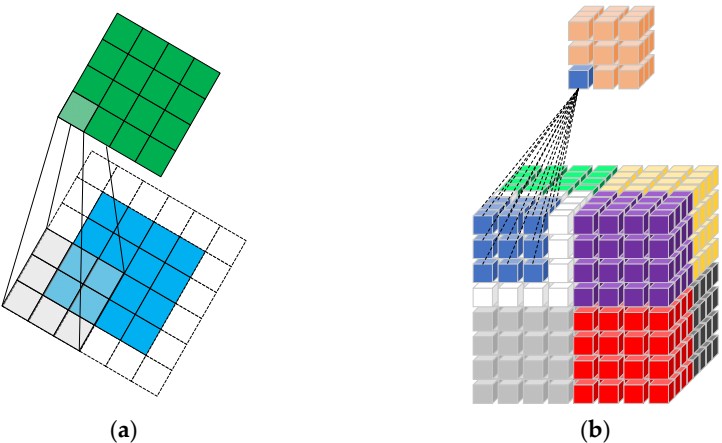

(**a**)　　　　　　　　　(**b**)

**Figure 3.** Schematic diagrams of (**a**) 2-D convolution and (**b**) 3-D convolution.

## 2.2. Framework of Proposed Method

The proposed Bi-LSTM-CNN network is based on the combination of Bi-LSTM and 3-D CNN for HSI classification. The method consists of two main parts—one for extracting spectral features through Bi-LSTM on the raw data and the other for extracting spatial-spectral features using 3-D CNN after the PCA dimension reduction on the data. To optimize the parameters of two subnetworks simultaneously, we concatenate the last of the FC layers of the Bi-LSTM and 3-D CNN to form a new FC layer, after which a softmax function is added. In this framework, the dimensionality of the raw data is decreased by the PCA to minimize the computational effort of 3-D convolution, and Bi-LSTM manages the original data to compensate for the spectral loss after dimensionality reduction. In addition, Bi-LSTM can better handle the contextual information of the spectra and fully exploit the spectral features of the HSI. After the last FC layer of each subnetwork, auxiliary functions are added to balance the contribution of the two subnetworks to the whole framework. The framework diagram of the Bi-LSTM-CNN is shown in Figure 4.

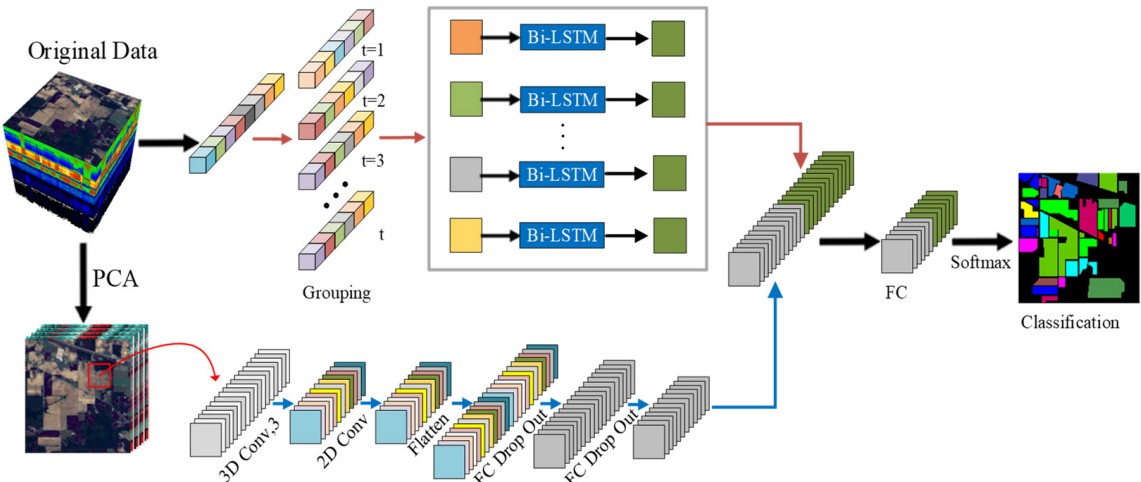

**Figure 4.** Framework diagram of the Bi-LSTM-CNN.

### 2.2.1. Bi-LSTM

In Section 2.1.1, we discussed the use of LSTM to process continuous HSI data and extract spectral information. The LSTM can retain only previous input information through cell states because it cannot access future input information. To handle this problem, we propose to extract spectral information using Bi-LSTM instead of LSTM. Unlike LSTM, Bi-LSTM preserves both latter and previous information. Additionally, Bi-LSTM shows accurate results with a better understanding of the context.

The Bi-LSTM network focuses on spectral contextual information. In general, the Bi-LSTM network inputs one band at a time step. However, HSI has hundreds of bands, making the Bi-LSTM network too deep to obtain a robust network under the condition of limited HSI samples. Therefore, the strategy used to group the spectral bands is crucial to improve HSI classification results. In Bi-LSTM, $t$ denotes the number of groups; $n$ denotes the number of bands; and $m = floor(n/t)$ represents the sequence length of each group, where $floor(x)$ denotes rounding down $x$. $Z = [Z_1, Z_2, \ldots Z_i, \ldots Z_n]$ is the spectral vectors per pixel in the HSI, where $Z_i$ is the reflectance value of the $i$th band. The grouping strategy is shown as Equation (9):

$$
\begin{aligned}
x^{(1)} &= \left[ Z_1, Z_{1+t}, \ldots Z_{1+t(m-1)} \right] \\
x^{(2)} &= \left[ Z_2, Z_{2+t}, \ldots Z_{2+t(m-1)} \right] \\
&\quad \ldots \\
x^{(i)} &= \left[ Z_i, Z_{i+t}, \ldots Z_{i+t(m-1)} \right] \\
&\quad \ldots \\
x^{(t)} &= \left[ Z_t, Z_{2t}, \ldots Z_{tm} \right],
\end{aligned}
\tag{9}
$$

where $x^{(i)}$ is the $i$th group. In this strategy, there will be a relative shortening of the spectral distance between different groups, and most of the spectral range will be covered.

The framework diagram of the Bi-LSTM is shown in Figure 5. The Bi-LSTM contains information about the forward and backward of the input sequence. At time $t$ of the input sequence, the forward LSTM layer contains information before time $t$, while the backward LSTM layer contains information after time $t$. The vectors output from the two LSTM layers are processed using concatenation. In Bi-LSTM, the colored squares represent each grouping. Each blue LSTM square represents an LSTM unit, and the red and green arrows indicate the forward LSTM and the backward LSTM, respectively, and the two LSTMs pass the information along the arrow direction. Meanwhile, the forward LSTM and the backward LSTM have the same input data.

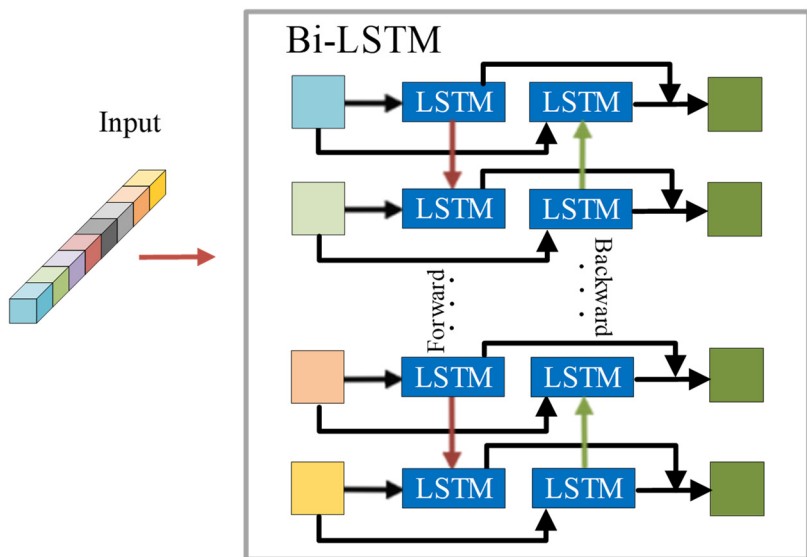

**Figure 5.** Bi-LSTM framework diagram.

### 2.2.1.1. 3-D CNN

The 3-D CNN has unique advantages in processing spatial-spectral features. Since there are many bands in an HSI, the 3-D convolution has a large computational complexity when extracting spatial-spectral features, which influences the efficiency of HSI classification. Therefore, in the Bi-LSTM-CNN, the 3-D CNN is used for the HSI after PCA dimensionality reduction to reduce the computational complexity.

HSI is denoted by $X \in R^{W \times H \times B}$, where $X$ represents the original data, $W$ and $H$ represent the width and the height, respectively, and $B$ denotes the number of spectral bands. Each HSI pixel in $X$ contains a one-hot label vector $Y = (y^1, y^2, y^3, \ldots, y^C) \in R^{1 \times 1 \times C}$ and a value in each of the $B$ spectral bands, where $C$ denotes the land-cover categories. During the convolution operation, to remove the spectral correlation and decrease the computational costs, the number of spectral bands is decreased from $B$ to $P$ by the PCA, while keeping the $W$ and $H$ of the spatial dimension constant. The spectral bands are reduced, but the essential spatial information for HSI classification is preserved. We create the 3-D patches centered on each pixel and extract adjacent regions of size $w \times w \times P$,

where $w \times w$ denotes the size of the window and $P$ denotes the number of first principal components that have been reserved by the PCA. The label of the central pixel decides the truth labels of the patches. The dataset is represented by $XP \in R^{M \times w \times w \times P}$, where $M$ represents the number of samples.

To achieve the spatial-spectral feature maps in the 3-D CNN, 3-D convolution is executed three times. Considering the crucial role of 2-D convolution in spatial information extraction, we apply 2-D convolution to increase the spatial feature maps before the flatten layer. To prevent overfitting and to improve the generalization of this model, the dropout is applied once after each FC layer. The structure of the 3-D CNN is shown in Figure 6.

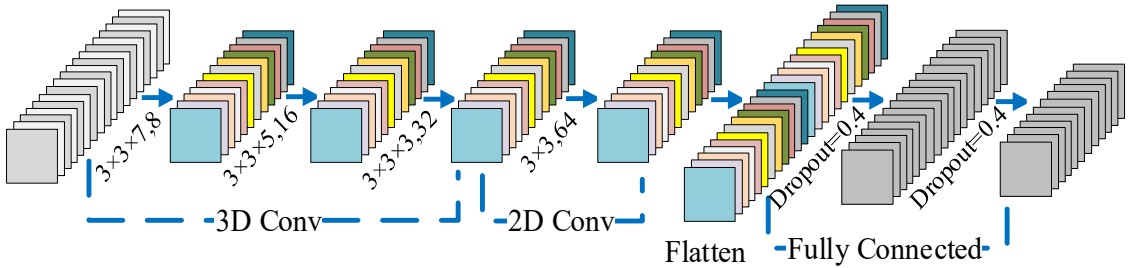

**Figure 6.** 3-D CNN framework diagram. Each convolutional layer is used with a ReLU function. The stride of each layer is 1.

### 2.2.2. Loss Function

In this framework, we adopt the softmax function after the final FC layer to determine the probability distributions over the pixel classes. In addition, to increase the nonlinearity and accelerate the convergence of the Bi-LSTM-CNN, we adopt the rectified linear units (ReLU) function after each layer.

To better train the parameters of the whole framework, after the final FC layer in the Bi-LSTM and the 3-D CNN, we adopt the auxiliary loss function. The complete loss function is defined as Equations (10)–(13):

$$L^{joint} = -\frac{1}{m} \sum_{i=1}^{m} \left[ y_i \log\left(\hat{y}_i^{joint}\right) + (1 - y_i) \log\left(1 - \hat{y}_i^{joint}\right) \right], \tag{10}$$

$$L^{Bi\text{-}LSTM} = -\frac{1}{m} \sum_{i=1}^{m} \left[ y_i \log\left(\hat{y}_i^{Bi\text{-}LSTM}\right) + (1 - y_i) \log\left(1 - \hat{y}_i^{Bi\text{-}LSTM}\right) \right], \tag{11}$$

$$L^{CNN} = -\frac{1}{m} \sum_{i=1}^{m} \left[ y_i \log\left(\hat{y}_i^{CNN}\right) + (1 - y_i) \log\left(1 - \hat{y}_i^{CNN}\right) \right], \tag{12}$$

$$L = L^{joint} + L^{Bi\text{-}LSTM} + L^{CNN}, \tag{13}$$

where $L$ represents the loss function, and $\hat{y}_i$ and $y_i$ refer to the predicted label and true label for the $i$th training sample, respectively. The superscript $joint$, $Bi\text{-}LSTM$, and $CNN$ denote the whole framework, the Bi-LSTM network, and the 3-D CNN, respectively. The variable $m$ denotes the number of the training sample. The parameters of the Bi-LSTM-CNN were optimized by the mini-batch stochastic gradient descent (SGD) algorithm.

The implementation procedure of the proposed Bi-LSTM-CNN method is shown in Algorithm 1.

| **Algorithm 1.** Bi-LSTM-CNN procedure | |
|---|---|
| **Input** | |
| | (1) HSI with labels. |
| | (2) The size of the patch $w$, the number of retained principal components $P$. |
| **Step 1** | For each pixel in the HSI, use Equation (9) to divide the hyperspectral cube into $t$ sequences as the input of the Bi-LSTM network. |
| **Step 2** | Retain the first $P$ principal components with PCA. Extract a patch of size $w \times w \times P$ in the neighborhood of each pixel after the reduced-dimensional HSI as the input of the 3-D CNN. |
| **Step 3** | Initialize the weights in the Bi-LSTM-CNN by assigning random values that follow a Gaussian distribution, where the mean, standard deviation, and bias terms are initialized to 0, 0.1, and 0, respectively. |
| **Step 4** | Import training samples into the Bi-LSTM-CNN. Bi-LSTM and 3-D CNN extract the spectral features and spatial-spectral features for HSI, respectively. Then, softmax is applied to classify the extracted features. Finally, the mini-batch SGD algorithm is exploited to optimize the parameters of the Bi-LSTM-CNN, and the parameters are adjusted by backpropagation to obtain the optimal parameters. |
| **Step 5** | For each pixel in the HSI, input the corresponding data from Step 1 and Step 2 to the Bi-LSTM-CNN to obtain the predicted value for the HSI. |
| **Output** | |
| | Prediction results for each pixel of HSI. |

## 3. Results

In this section, three open HSI datasets (the three datasets are available at http://www.ehu.eus/ccwintco/index.php/Hyperspectral_Remote_Sensing_Scenes accessed on 14 November 2020) are evaluated in the performance of the Bi-LSTM-CNN by applying three evaluation metrics, comprising overall accuracy (OA), average accuracy (AA), and kappa coefficient (Kappa).

### 3.1. Dataset Description and Training Details

The Indian Pines (IP) is an image acquired by the AVIRIS sensor in 1992 from the Agricultural and Forestry Hyperspectral Experiment site in northwestern Indiana. The dataset is an agricultural region with geometrically regular crop areas but irregular forest areas. The dataset is the size of 145 × 145 and has 224 spectral reflectance bands. The spatial resolution of each pixel is 20 m. Of these, 24 spectral bands were excluded because they covered the water absorption region; the wavelength range of the retained 200 bands is 0.4–2.5 μm. The available samples were divided into 16 categories, representing approximately half of the total data. The false-color composite image and the ground-truth map correspond to in Figure 7a,b respectively. In the experiment, the dataset was selected randomly in the labeled parts from each category, and the ratio of the training and testing set was 1:9. Table 1 exhibits the details of the sample, as well as the corresponding colors of the ground-truth map.

The University of Pavia (PU) campus in northern Italy was gathered by the ROSIS-03 sensor in 2001. This scene has a size of 610 × 340 × 115 and a wavelength range of 0.43–0.86 μm. The spatial resolution of each pixel is 1.3 m. This scene contained nine categories and 103 spectral bands after 12 noisy bands were discarded. The false-color composite image and the ground-truth map correspond to in Figure 8a,b respectively. Table 2 exhibits the details of the sample, as well as the corresponding colors of the ground-truth map. In the labeled pixels from the PU, only 5% were used as the training set and the rest as the testing set.

The Salinas Valley (SV) scene is an image of the Salinas Valley, California, collected by the AVIRIS sensor. This scene forms a cube of dimension 512 × 217 × 224, and the spatial resolution of each pixel is 3.7 m. Similar to the IP dataset, after discarding 20 water absorption and noise bands, only 204 bands remained. This scene included 16 different agriculture crop categories. The false-color composite image and the ground-truth map correspond to in Figure 9a,b respectively. Table 3 exhibits the details of the sample, as well as the corresponding colors of the ground-truth map. Among the labeled pixels of this

scene, only 5% were used as the training set and the rest as the testing set. In the above three datasets, the data of all training sets are randomly selected.

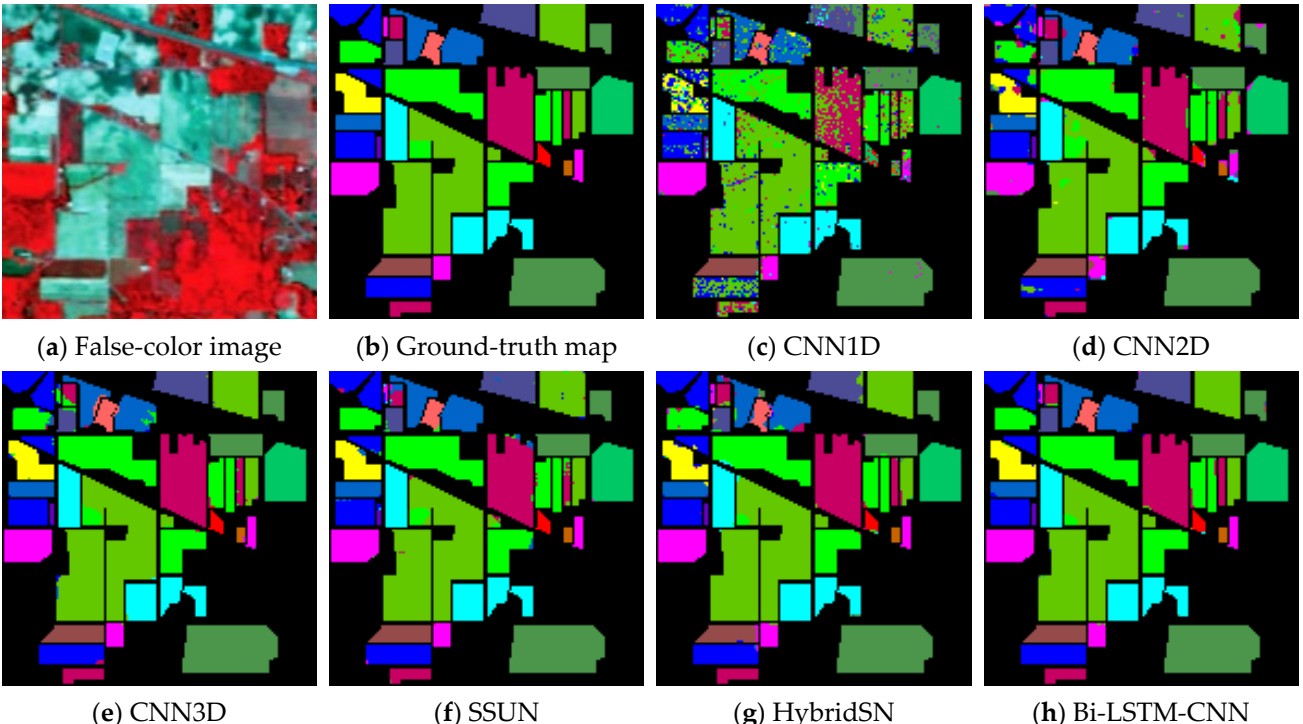

**Figure 7.** Classification maps for the IP dataset with 10% of training data. (**a**) False-color image, (**b**) Ground-truth map, (**c**) CNN1D, (**d**) CNN2D, (**e**) CNN3D, (**f**) SSUN, (**g**) HybridSN, (**h**) Bi-LSTM-CNN.

**Table 1.** Details of the IP dataset.

| Class NO. | Class Name | Color | Number of Training Samples | Number of Testing Samples | Total Number of Samples |
|---|---|---|---|---|---|
| C1 | Alfalfa | | 5 | 41 | 46 |
| C2 | Corn—no till | | 143 | 1285 | 1428 |
| C3 | Corn—min till | | 83 | 747 | 830 |
| C4 | Corn | | 24 | 213 | 237 |
| C5 | Grass/pasture | | 48 | 435 | 483 |
| C6 | Grass/tree | | 73 | 657 | 730 |
| C7 | Grass/pasture—mowed | | 3 | 25 | 28 |
| C8 | Hay—windrowed | | 48 | 430 | 478 |
| C9 | Oats | | 2 | 18 | 20 |
| C10 | Soybeans—no till | | 97 | 875 | 972 |
| C11 | Soybeans—min till | | 245 | 2210 | 2455 |
| C12 | Soybeans—clean till | | 59 | 534 | 593 |
| C13 | Wheat | | 20 | 185 | 205 |
| C14 | Woods | | 126 | 1139 | 1265 |
| C15 | Bldg-grass-tree-drives | | 39 | 347 | 386 |
| C16 | Stone-steel towers | | 9 | 84 | 93 |
| | Total | | 1024 | 9225 | 10,249 |

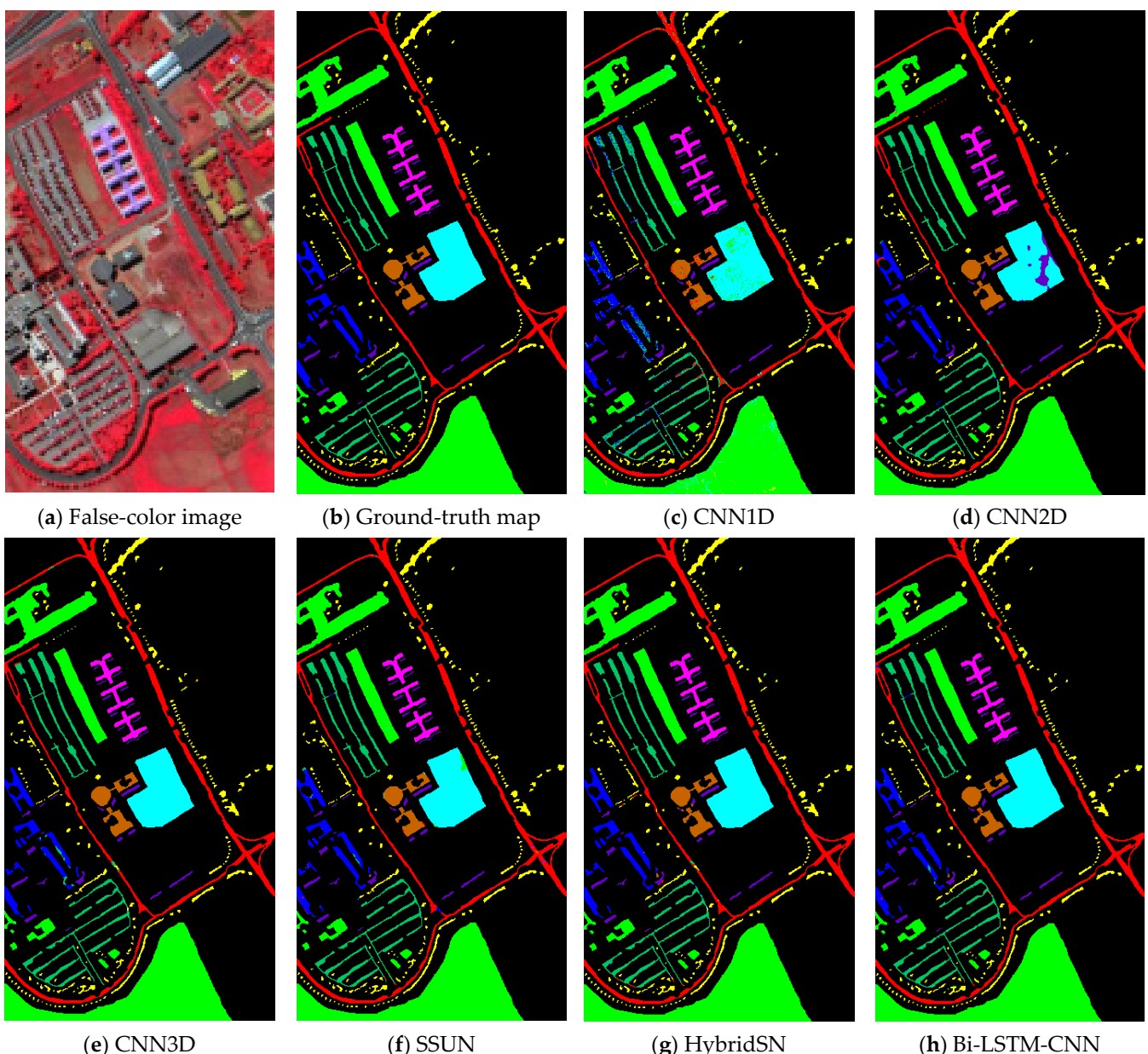

**Figure 8.** Classification maps for the PU dataset with 10% of training data. (**a**) False-color image; (**b**) Ground-truth map; (**c**) CNN1D; (**d**) CNN2D; (**e**) CNN3D; (**f**) SSUN; (**g**) HybridSN; (**h**) Bi-LSTM-CNN.

**Table 2.** Details of the PU dataset.

| Class NO. | Class Name | Color | Number of Training Samples | Number of Testing Samples | Total Number of Samples |
|---|---|---|---|---|---|
| C1 | Asphalt | | 332 | 6299 | 6631 |
| C2 | Meadows | | 933 | 17,716 | 18,649 |
| C3 | Gravel | | 105 | 1994 | 2099 |
| C4 | Trees | | 153 | 2911 | 3064 |
| C5 | Painted metal sheets | | 67 | 1278 | 1345 |
| C6 | Bare soil | | 251 | 4778 | 5029 |
| C7 | Bitumen | | 67 | 1263 | 1330 |
| C8 | Self-blocking bricks | | 184 | 3498 | 3682 |
| C9 | Shadows | | 47 | 900 | 947 |
| Total | | | 2139 | 40,637 | 42,776 |

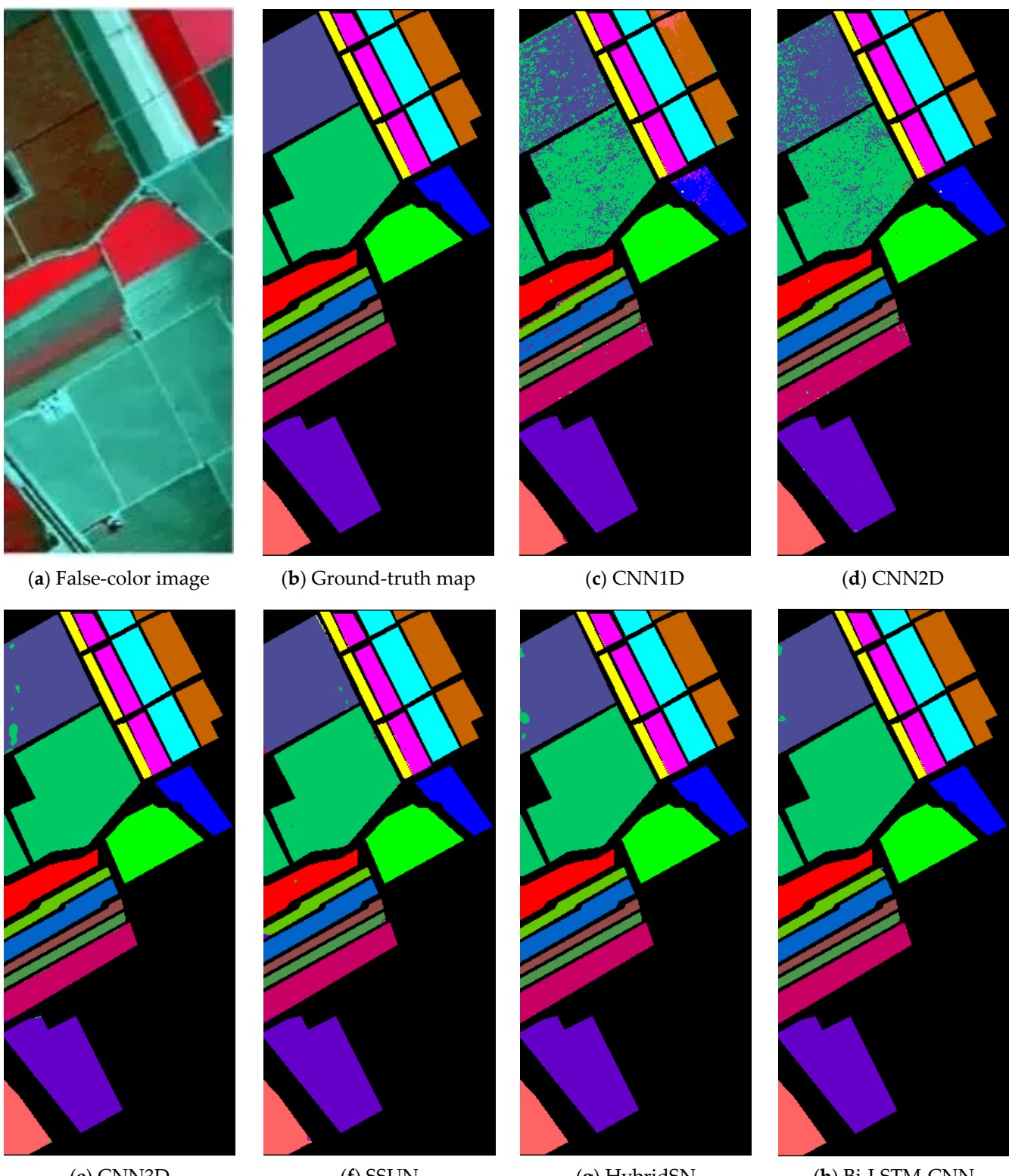

Figure 9. Classification maps for the SV dataset with 10% of training data. (**a**) False-color image; (**b**) Ground-truth map; (**c**) CNN1D; (**d**) CNN2D; (**e**) CNN3D; (**f**) SSUN; (**g**) HybridSN; (**h**) Bi-LSTM-CNN.

**Table 3.** Details of the SV dataset.

| Class NO. | Class Name | Color | Number of Training Samples | Number of Testing Samples | Total Number of Samples |
|---|---|---|---|---|---|
| C1 | Brocoli_green_weeds_1 | | 100 | 1909 | 2009 |
| C2 | Brocoli_green_weeds_2 | | 186 | 3540 | 3726 |
| C3 | Fallow | | 99 | 1877 | 1976 |
| C4 | Fallow_rough_plow | | 70 | 1324 | 1394 |
| C5 | Fallow_smooth | | 134 | 2544 | 2678 |
| C6 | Stubble | | 198 | 3761 | 3959 |
| C7 | Celery | | 179 | 3400 | 3579 |
| C8 | Grapes_untrained | | 564 | 10,707 | 11,271 |
| C9 | Soil_vinyard_develop | | 310 | 5893 | 6203 |
| C10 | Corn_senesced_green_weeds | | 164 | 3114 | 3278 |
| C11 | Lettuce_romaine_4wk | | 53 | 1015 | 1068 |
| C12 | Lettuce_romaine_5wk | | 96 | 1831 | 1927 |
| C13 | Lettuce_romaine_6wk | | 46 | 870 | 916 |
| C14 | Lettuce_romaine_7wk | | 54 | 1016 | 1070 |
| C15 | Vinyard_untrained | | 364 | 6904 | 7268 |
| C16 | Vinyard_vertical_trellis | | 90 | 1717 | 1807 |
| | Total | | 2707 | 51,422 | 54,129 |

*3.2. Experimental Configuration*

All the experiments were run with an NVIDIA GTX 1070 GPU and an Inter i7-6700 3.40-GHz CPU with 32 GB of RAM. We performed randomized training and test data replication 10 times for each test. Based on several experiments, we chose 0.0001 as the best learning rate. To optimize the Bi-LSTM-CNN, we adopted the mini-batch SGD algorithm, with a batch size of 128. In the Bi-LSTM, the spectral bands are divided into three groups.

In Table 4, it is evident that the Bi-LSTM-CNN obtained the optimal results when the input window of the 3-D patches was $25 \times 25$. Table 5 shows the effect of $P$ on the classification results in the IP dataset. When $P$ is 30, the classification results are best. If $P$ keeps increasing, the number of network parameters will increase sharply. Therefore, the input size of the 3-D patches was $25 \times 25 \times 30$. In Figure 10, the curves of classification accuracy with epochs during training over IP, PU, and SV datasets. When the epoch is 100, the classification accuracy was close to 1, but there was still instability. The classification accuracy was stable when the epoch reached 300, so the epoch of the Bi-LSTM-CNN was adopted 300. The parameters for the proposed Bi-LSTM-CNN method on the IP dataset are shown in Table 6.

**Table 4.** Impact of the input window size of the 3-D patches on the performance.

| Window | IP | | | PU | | | SV | | |
|---|---|---|---|---|---|---|---|---|---|
| | OA (%) | AA (%) | Kappa $\times$ 100 | OA (%) | AA (%) | Kappa $\times$ 100 | OA (%) | AA (%) | Kappa $\times$ 100 |
| $10 \times 10$ | 95.41 | 93.07 | 94.86 | 96.35 | 95.18 | 95.17 | 95.01 | 97.57 | 94.45 |
| $15 \times 15$ | 97.93 | 96.11 | 97.65 | 98.15 | 97.39 | 97.55 | 97.78 | 98.91 | 97.53 |
| $20 \times 20$ | 98.29 | 97.60 | 98.04 | 99.06 | 98.42 | 98.76 | 99.14 | 99.56 | 99.04 |
| $25 \times 25$ | 98.63 | 98.50 | 98.45 | 99.56 | 99.22 | 99.42 | 99.81 | 99.83 | 99.84 |

**Table 5.** In the IP dataset, impact of the number of retained principal component on the performance.

| $P$ | OA (%) | AA (%) | Kappa $\times$ 100 | $P$ | OA (%) | AA (%) | Kappa $\times$ 100 |
|---|---|---|---|---|---|---|---|
| 15 | 98.13 | 97.77 | 97.86 | 25 | 98.55 | 98.46 | 98.35 |
| 20 | 98.18 | 97.87 | 97.92 | 30 | 98.64 | 98.50 | 98.45 |

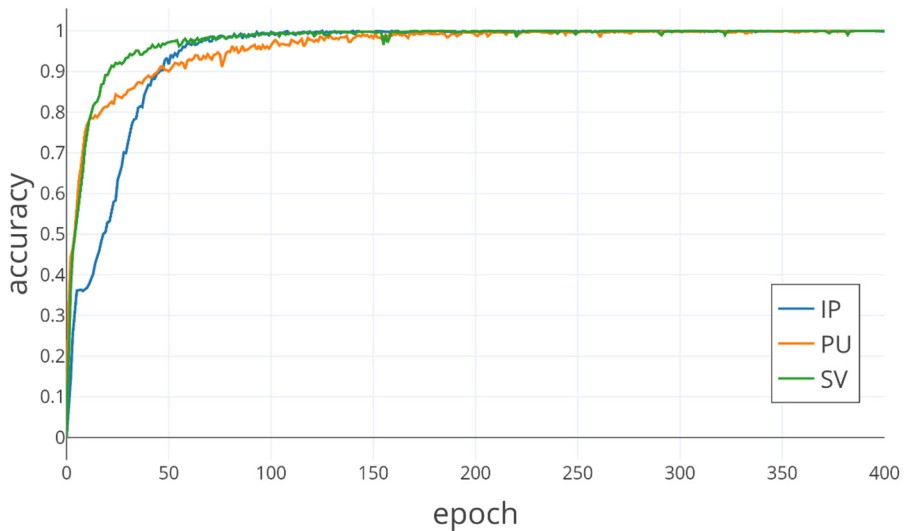

**Figure 10.** In three datasets, classification accuracy for each epoch during training.

**Table 6.** Parameters for the proposed Bi-LSTM-CNN method on the IP dataset.

| Layer (Type) | Output Shape | Param # |
| --- | --- | --- |
| CNNInput (InputLayer) | (25, 25, 30, 1) | 0 |
| conv3d1 (Conv3D) | (23, 23, 24, 8) | 512 |
| conv3d2 (Conv3D) | (21, 21, 20, 16) | 5776 |
| conv3d3 (Conv3D) | (19, 19, 18, 32) | 13856 |
| reshape1 (Reshape) | (19, 19, 576) | 0 |
| conv2d1 (Conv2D) | (17, 17, 64) | 331840 |
| flatten1 (Flatten) | (18496) | 0 |
| dense1 (Dense) | (256) | 4735232 |
| BiLSTMInput (InputLayer) | (3, 66) | 0 |
| dropout1 (Dropout) | (256) | 0 |
| bidirectional1 (Bidirectional) | (256) | 199680 |
| dense2 (Dense) | (128) | 32896 |
| BiLSTMDense (Dense) | (128) | 32896 |
| dropout2 (Dropout) | (128) | 0 |
| concatenate1 (Concatenate) | (256) | 0 |
| JOINTDENSE (Dense) | (128) | 32896 |
| JOINTSOFTMAX (Dense) | (16) | 2064 |
| LSTMSOFTMAX (Dense) | (16) | 2064 |
| CNNSOFTMAX (Dense) | (16) | 2064 |
| Total params #: 5,391,776 | | |

### 3.3. Classification Results

In this paper, we compared the Bi-LSTM-CNN with some state-of-the-art methods, which are CNN1D, CNN2D, and CNN3D [56], SSUN [57], and HybridSN [58]. CNN1D, CNN2D, and CNN3D used 1-D convolution, 2-D convolution, and 3-D convolution, respectively. SSUN used an LSTM and a multiscale CNN to extract spectral and spatial features for implementing spatial and spectral joints. HybridSN adopted a mixture of 3-D convolution and 2-D convolution to extract spatial-spectral features with mainly spatial information. All the comparison methods were run in the same environment.

Tables 7–9 show the results acquired by six methods on the IP (10% of the total dataset), PU (5%), and SV (5%), including OA, AA, Kappa, testing time, accuracy for each class. These are the result of running on the testing set. CNN1D had the worst classification results. In detail, all three evaluation metrics (OA, AA, and Kappa) of CNN1D were

lower than the other methods. The accuracy of each class is the lowest among the six methods. CNN2D had better classification results than CNN1D, but still had a large drawback with other methods. Among the remaining methods, each method achieves the best results in some classes. Specifically, the Bi-LSTM-CNN obtained higher performance than other methods on OA, AA, Kappa. In addition, the Bi-LSTM-CNN obtains the highest accuracy in most of classes. The testing time of CNN1D, CNN2D, SSUN are less than other comparison methods.

**Table 7.** Classification results for the IP dataset using 10% of the available labeled data.

| Class NO. | CNN1D | CNN2D | CNN3D | SSUN | HybridSN | Bi-LSTM-CNN |
|-----------|-------|-------|-------|------|----------|-------------|
| C1 | 32.44 | 69.51 | 80.73 | **99.51** | 86.19 | 96.10 |
| C2 | 69.84 | 91.18 | **97.68** | 96.93 | 94.63 | 94.93 |
| C3 | 62.82 | 90.98 | 98.70 | 98.34 | **99.26** | 98.90 |
| C4 | 43.05 | 87.18 | 94.08 | 97.51 | 96.32 | **97.93** |
| C5 | 88.48 | 89.98 | 96.25 | 96.99 | **99.04** | 98.99 |
| C6 | 96.59 | 97.41 | **99.53** | 98.87 | 98.93 | 99.30 |
| C7 | 40.40 | 87.20 | 88.00 | 97.20 | **100.00** | **100.00** |
| C8 | 98.28 | 98.84 | 99.98 | 99.88 | 99.30 | **100.00** |
| C9 | 39.44 | 57.22 | 85.56 | 78.33 | 90.74 | **96.11** |
| C10 | 67.34 | 93.34 | 97.12 | 95.94 | 98.61 | **99.15** |
| C11 | 82.92 | 95.36 | 98.90 | 98.48 | 99.00 | **99.50** |
| C12 | 73.15 | 88.31 | 94.40 | 97.08 | 95.82 | **98.78** |
| C13 | 97.95 | 97.24 | **99.78** | 98.43 | 97.57 | 99.72 |
| C14 | 94.04 | 98.64 | 99.72 | 98.90 | 99.77 | **99.82** |
| C15 | 60.98 | 91.59 | 95.76 | 97.58 | 98.80 | **99.79** |
| C16 | 83.45 | 90.12 | 96.55 | 93.81 | 91.67 | **97.61** |
| OA (%) | 78.89 | 93.73 | 97.99 | 97.89 | 98.05 | **98.63** |
| AA (%) | 71.58 | 89.01 | 95.17 | 96.53 | 96.60 | **98.50** |
| Kappa × 100 | 74.88 | 92.85 | 97.70 | 97.60 | 97.78 | **98.45** |
| Testing time (s) | 0.36 | 0.61 | 3.49 | 1.19 | 4.54 | 5.34 |

**Table 8.** Classification results for the PU dataset using 5% of the available labeled data.

| Class NO. | CNN1D | CNN2D | CNN3D | SSUN | HybridSN | Bi-LSTM-CNN |
|-----------|-------|-------|-------|------|----------|-------------|
| C1 | 93.22 | 99.43 | 99.02 | 98.55 | **99.59** | **99.59** |
| C2 | 98.01 | 99.83 | 99.99 | 99.84 | 99.97 | **100.00** |
| C3 | 77.99 | 97.44 | 95.39 | 95.50 | **97.77** | 97.19 |
| C4 | 92.33 | 97.49 | 99.04 | 99.16 | 98.63 | **99.29** |
| C5 | 99.57 | 99.84 | 99.99 | 99.63 | 99.40 | **100.00** |
| C6 | 87.53 | 88.78 | **100.00** | 99.51 | 99.97 | **100.00** |
| C7 | 87.24 | 99.29 | **99.99** | 96.12 | 99.94 | 98.97 |
| C8 | 84.07 | 97.49 | 98.19 | **98.62** | 97.96 | 98.28 |
| C9 | 99.66 | 95.89 | 98.88 | 98.72 | 94.32 | **99.67** |
| OA (%) | 93.20 | 98.05 | 99.48 | 99.09 | 99.38 | **99.56** |
| AA (%) | 91.07 | 97.50 | 99.16 | 98.41 | 98.61 | **99.22** |
| Kappa × 100 | 90.94 | 97.42 | 99.31 | 98.79 | 99.19 | **99.42** |
| Testing time (s) | 1.41 | 3.40 | 14.63 | 4.45 | 7.29 | 10.62 |

**Table 9.** Classification results for the SV dataset using 5% of the available labeled data.

| Class NO. | CNN1D | CNN2D | CNN3D | SSUN | HybridSN | Bi-LSTM-CNN |
|---|---|---|---|---|---|---|
| C1 | 99.20 | 99.42 | **100.00** | 99.83 | **100.00** | **100.00** |
| C2 | 99.96 | 99.87 | **100.00** | 99.93 | **100.00** | **100.00** |
| C3 | 98.97 | 99.68 | **100.00** | 99.89 | **100.00** | **100.00** |
| C4 | 99.20 | 99.31 | 99.66 | 99.68 | **99.99** | **99.99** |
| C5 | 97.87 | 98.56 | **99.87** | 99.39 | 98.15 | 99.76 |
| C6 | 99.80 | 99.68 | **100.00** | 99.99 | **100.00** | **100.00** |
| C7 | 98.66 | 99.09 | 99.96 | 99.97 | 99.99 | **100.00** |
| C8 | 82.07 | 90.98 | 99.81 | 99.58 | 99.94 | **99.96** |
| C9 | 99.66 | 99.80 | 99.97 | 99.99 | **100.00** | **100.00** |
| C10 | 95.86 | 98.60 | 99.65 | 99.73 | 99.64 | **99.97** |
| C11 | 97.68 | 98.88 | 99.07 | 99.11 | 98.72 | **99.31** |
| C12 | 99.89 | 99.63 | 99.89 | **100.00** | 99.98 | **100.00** |
| C13 | 98.60 | 99.01 | 99.88 | 99.43 | **100.00** | 99.89 |
| C14 | 96.16 | 97.17 | 99.46 | 98.13 | 99.50 | **99.51** |
| C15 | 76.71 | 91.34 | 98.30 | 99.09 | 98.20 | **99.30** |
| C16 | 98.34 | 98.43 | 99.77 | 99.36 | **99.99** | 99.94 |
| OA (%) | 92.41 | 95.01 | 99.67 | 99.63 | 99.71 | **99.81** |
| AA (%) | 96.23 | 96.87 | 99.77 | 99.57 | 99.77 | **99.83** |
| Kappa × 100 | 91.56 | 94.86 | 99.63 | 99.59 | 99.67 | **99.84** |
| Testing time (s) | 2.34 | 2.99 | 18.15 | 5.70 | 9.40 | 13.16 |

The classification maps of the six methods in the three datasets are shown in Figures 7–9. These Figures show the prediction results of the six methods for all labeled samples. It is obvious that the classification maps of CNN1D and CNN2D have a large amount of salt and pepper noise. As the remaining four methods used spatial and spectral information, the classification map approximates more closely to the ground-truth map. In particular, the Bi-LSTM-CNN has very few pixel points that are different from the ground-truth map.

## 4. Discussion

In the experiment result, it is obvious that the Bi-LSTM-CNN significantly outperforms the other methods. The OA of the CNN1D method did not exceed 94% in all the considered datasets. Since the input data of CNN1D is a 1-D vector, spatial information of the input data is lost, resulting in the worst classification results of CNN1D among all methods. The CNN2D model considers the spatial information, which makes the classification results an improvement compared to CNN1D. Thus, it shows that spatial information is critical for HSI classification.

However, the CNN2D model has problems, which usually result in degraded shapes of some objects and materials. The union of spatial and spectral information can address this issue, and the other methods (CNN3D, SSUN, HybridSN, and Bi-LSTM-CNN) all achieve more similar classification results to the corresponding ground-truth maps. The SSUN model extracts spatial and spectral features separately, which are integrated and then sent to the classifier for classification. As spatial features dominate the classification results, SSUN is unable to effectively balance the two features, thus resulting in a little contribution of spectral features to the classification results. The CNN3D model directly extracts the spatial-spectral features of the HSI, but to decline the computational complexity of the convolutional layers, the PCA dimensionality reduction is performed on the input data. Hence, a small amount of spectral information is lost. Despite this, CNN3D still spends a lot of time in the testing phase on the PU and SV dataset compared to HybridSN and Bi-LSTM-CNN.

The HybridSN model replaces the final 3-D convolutional layer with 2-D convolution, decreasing the number of parameters in the network while maintaining accuracy. However,

in the PU dataset experiments, the OA of the HybridSN model is lower than the CNN3D model, and the generalizability of the HybridSN method is slightly worse. In the Bi-LSTM-CNN, the lack of 3-D CNN processing spectral information is compensated, and the experimental results after adding Bi-LSTM are significantly better than the other methods.

In the classes with a small number of samples, the Bi-LSTM-CNN method also obtains better classification results. In the IP dataset, due to the very small number of labeled samples in some classes, the number of available training samples is extremely small. For example, the number of samples for C1, C7, C9, C16 is not more than ten, which greatly increases the learning difficulty for these classes. Except for C1, the best classification accuracy is obtained for several other categories. Except for C1, the Bi-LSTM-CNN method obtains a higher OA in the other classes than other methods. In the PU and SV datasets, the number of training samples for each class is sufficient for the Bi-LSTM-CNN method, although there is a large difference in the number of samples for different classes.

## 5. Conclusions

This paper proposed a unified network framework that contained a band-grouping-based Bi-LSTM network and a 3-D CNN for HSI classification. In this network, Bi-LSTM can extract high-quality spectral features considering complete spectral contextual information, which compensates for the shortcomings of the 3-D CNN. The Bi-LSTM-CNN network is able to harness the strengths of both subnetworks by using auxiliary loss functions. Compared with the model using only 3-D CNN, the Bi-LSTM-CNN can obtain better classification results by adding a few parameters. In the PU and SV datasets, we validated the performance of the model using less training data (5%). The experimental results showed that the Bi-LSTM-CNN method significantly improved the accuracy of HSI classification. In future work, we will either replace the LSTM with the Gated Recurrent Unit to improve the speed of the network or use the optimized 3-D CNN to further improve the HSI classification results.

**Author Contributions:** Methodology, software and conceptualization, J.Y. and Q.C.; modification and writing—review and editing, C.Q.; investigation and data curation, J.Q. All authors have read and agreed to the published version of the manuscript.

**Funding:** This research was funded by the Henan Province Science and Technology Breakthrough Project, grant number 212102210102 and 212102210105.

**Acknowledgments:** The authors would like to thank the editors and reviewers for their advice.

**Conflicts of Interest:** The authors declare no conflict of interest.

## Abbreviations

The abbreviations in this paper are as follows:

| | |
|---|---|
| HSI | Hyperspectral Image |
| Bi-LSTM | Biderectional Long Short-Term Memory |
| CNN | Convolutional Neural Network |
| SVM | Support Vector Machine |
| KNN | K-Nearest Neighbor |
| SAE | Stacked Auto-Encoder |
| DBN | Deep Belief Network |
| RNN | Recurrent Neural Network |
| GAN | Generative Adversarial Network |
| AE | Auto-Encoder |
| PCA | Principal Component Analysis |
| FC | Fully Connected |
| LSTM | Long Short-Term Memory |
| ReLU | Rectified Linear Unit |
| OA | Overall Accuracy |

| AA | Average Accuracy |
| Kappa | Kappa Coefficient |
| PC | Principal Component |
| IP | Indian Pines |
| PU | University of Pavia |

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
