# Peer review of "Spatial-Spectral Network for Hyperspectral Image Classification: A 3-D CNN and Bi-LSTM Framework"

_remotesensing, doi:10.3390/rs13122353_

Round 1

Reviewer 1 Report

Please tak into accont the comments that I indicated as notes in the attached PDF

Reviewer 2 Report

This work describes a Spatial-Spectral Network for Hyperspectral Image Classification: A 3-D CNN and Bi-LSTM Framework.

Firstly, the state-of-the-art description is lacking references to Visual Transformers and other than PCS, e.g. MAF.

There is no reason to explain RNN when it is not used.

Notation: It seems that tanh and Ø is used for the same function.

It is stated that CNN always contain convolutional and pooling layers and an activation function, which is not always necessarily the case.

The division and use of test, train and validation data should be clearly explained. It is also unclear whether the results presented are for training data, as no validation date is mentioned.

Why is only 5% of the data used for training?

How can you get 100% accuracy? This is usually a sign for severe overfitting or something is wrong.

Reviewer 3 Report

I list here below some comments to improve the presentation, but in general I would say that the manuscript requires minor revisions only. 

1. Introduction
LL.48-51
Do you mean the kernel trick? Please clarify which kernels have been used.

LL.80-82
There are some dimensionality reduction methods as well as PCA and KPCA. More literature review is required.

Hyperparameters of 3-D CNN
Why did you choose these configurations?

4. Discussion
The discussion aspect is weak and needs to be improved. 
What are the impacts of training data characteristics (e.g. class imbalance) on the performance of 3-D CNN?

Round 2

Reviewer 2 Report

As to point 1: can you comment on the use of PCA and why this should be better than transformers?

As to point 5: Please show the distribution of classes before the split. This is see potential unbalanced classes, which can have severe effects.

As to point 7: It is still very suspicious with 100% accuracy. This might also be related to class imbalance.
